

# Global warming causes sinkhole collapse – Case study in Florida, USA

Yan Meng [1,2,*], Long Jia [2]

[1]School of environmental studies, China university of Geosciences, Wuhan, 430074,China
[2]Institute of Karst Geology, Chinese Academy of Geological Sciences/Karst Dynamics Laboratory, MLR & GZAR, CGS, Guilin, 541004,China

*Correspondence to*: Yan Meng (sinkhole@163.com)

**Abstract.** The occurrence frequency and intensity of many natural geohazards, such as landslides, debris flows and earthquakes, have increased in response to global warming. However, the effects of such on development and spread of
sinkholes has been largely overlooked. Most research shows that water pumping and related drawdown is the most important factor in sinkhole development, but in this paper evidence is presented which highlights the role played by global warming in causing sinkholes. The state of Florida, USA is used as a case study in which the role of global warming is evident, based on correlation analysis between sinkhole collapse and peak drought periods. Three distinct drought and sinkhole collapse phases are evident between 1965 and 2006, along with eight peak periods of sinkhole collapse that lag slightly behind eight peak
drought periods. A prediction equation is derived according to curve fitting and a correlation coefficient of 0.999 is determined. The results of this study confirm that global warming related to climate change has led to an increase in sinkhole collapse events in Florida over the past 50 years, which is of significance for studying the occurrence and prediction of other sinkhole collapse events and global warming on an international scale.

**Keywords**
Geohazard; Drought; Karst; Trend prediction; Curve fitting

## 1 Introduction

Global warming resulting from climate change has altered the occurrence frequency and intensity of many natural geohazards, including landslides, debris flows and earthquakes (Calbó et al., 2010; Coe and Godt., 2012; Seneviratne et al., 2012; Gariano and Guzzetti., 2016; Heuvel, et al., 2016; Turkington, et al., 2016; Yongming Lin, et al., 2017). As an
example of the mechanism for this, research has shown that 5%−10% of global permafrost will melt if global temperatures rise by 2 ℃, causing a significant increase in landslides and mudslides (Dong and Jia., 2004).

Sinkholes are a widespread type of geohazard, mainly distributed in the United States, China, Italy, Spain and Russia (Gutiérrez, et al., 2014; Lei, et al., 2015). The impact of climate change on sinkhole occurrence is expected, because rising temperatures will change natural hydrological processes (Gabriella Szépszó, et al., 2014), enhance dissolution of limestone
(Mulec and Prelovšek., 2014) and promote soil failure (Zhou, et al., 2014). Recent reviews in the literature have shown that



sinkhole hazards will probably intensify in the future as a result of climate change (Rogelio Linares, et al., 2017), but quantification of the impact on sinkholes has been limited. This is largely because of a lack of long-term hydrological and climate data, and a lack of representative sinkhole inventories, inclusive of chronological information.

In this paper, the causal effects of global warming on sinkhole development and intensification are fully investigated using statistical analysis of sinkhole cases in the state of Florida, USA. In general, it can be shown that for every 0.1 ℃ rise in global temperature, the number of sinkholes increases by 1%−3%.

## 2 Materials and methods

### 2.1 Global temperature

Global warming as a result of climate change is a quantifiable phenomenon (Shi et al., 2010; Gariano, et al., 2016; Turkington, et al., 2016), with a demonstrable increase in global temperatures by ~0.57 ℃ over the last century (Fig. 1). It has been reported that the global surface temperature is likely to rise a further 0.3 to 1.7 ℃ in the lowest emissions scenario during the 21st century, or by 2.6 to 4.8 ℃ in the highest emissions scenario (IPCC, 2013).

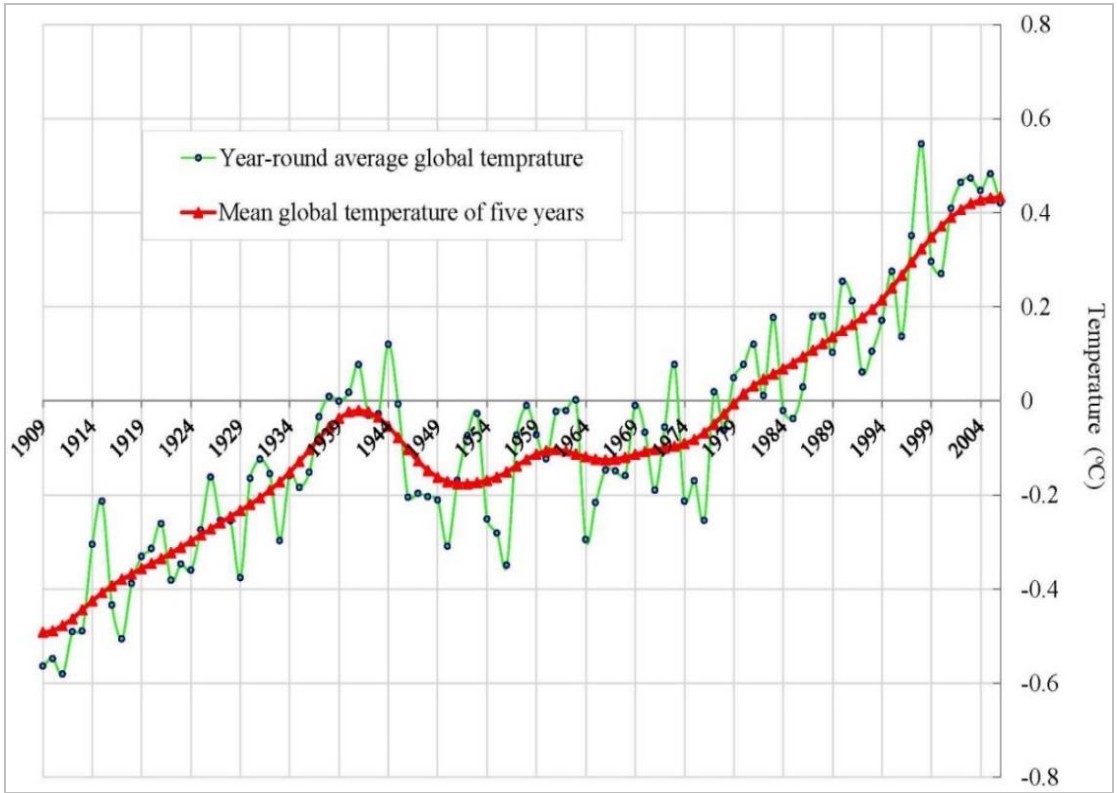

**Figure 1: Global mean surface-temperature changes from 1909 to 2006. The green line is the global annual mean, and the red line is the five-year local regression line (based on http://en.wikipedia.org/wiki/Global_warming).**



## 2.2 Sinkhole collapse events in Florida

In Florida, sinkhole collapse events are recorded in the Florida Subsidence Incident Report, authored by the Florida Geological Survey, which provides a primary publicly available sinkhole database. More than 2800 sinkholes have been reported in Florida since the 1950s, and 2786 of them were fully recorded between 1949 and 2006. The data recorded

includes occurrence time, location, shape, dimensions, soil type, side slope, land use and land cover (Han, et al., 2016). The long-term and complete records of such sinkholes form the basis of the time peak relationship analysis between sinkhole collapse and global warming.

The sinkhole collapses recorded in Florida have three distinct peaks (Fig. 2) and provide ideal research candidates, which is why this region was chosen for the study. The study is based on the 2786 sinkholes that have been well-documented.

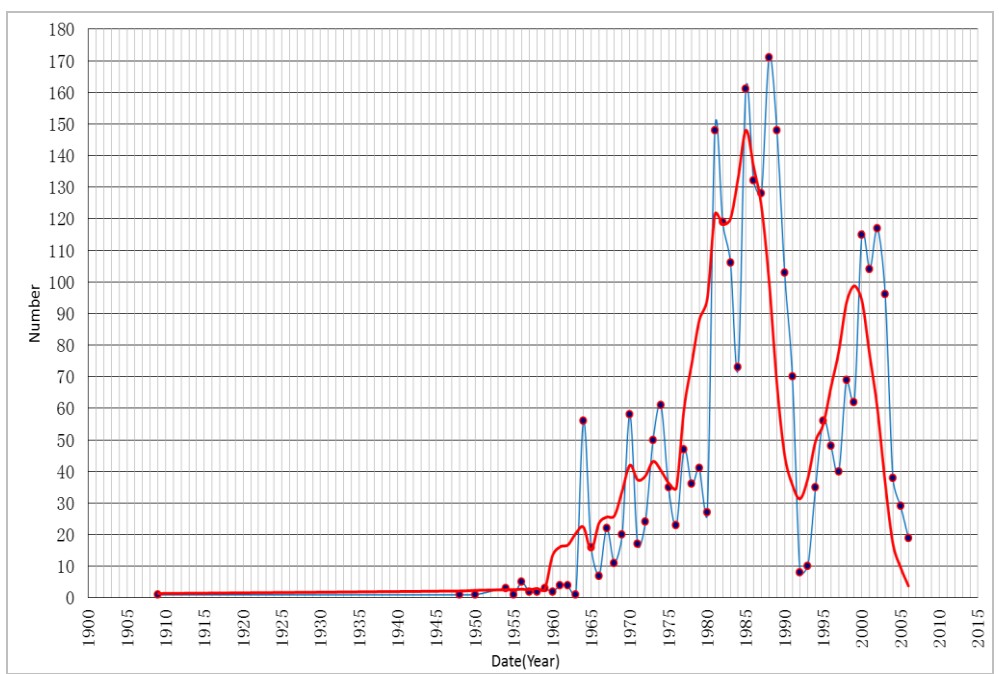


**Figure 2: Frequency of sinkhole formation between 1948 and 2006 in Florida, USA. The red line is the five-year local regression line (data obtained from the Florida Office of Insurance Regulation).**

Most research has shown that pumping of water and associated drawdown is the leading cause of sinkhole formation and collapse (Anikeev and Leonenko., 2014; Youssef, et al., 2016; Rogelio Linares, et al., 2017). However, the impact of global

warming should not be ignored. For example, altered global rainfall patterns and increasing evaporation because of higher temperatures leads to a decrease in groundwater flow, resulting in sinkhole formation, or such decreased flow may lead to intensification of water pumping and related drawdown in urban and industrial areas that in itself leads to groundwater level reduction and related sinkhole development.

Also, the addition of greenhouse gases to the atmosphere and global warming increase the dissolution of bedrock, thus

increasing the intensity and frequency of sinkhole collapse. This is especially true for areas underlain by limestone or



dolomite, in which the basic carbonate dissolution formula $CaCO_3 + 2H^+ \rightarrow Ca^{2+} + H_2O + CO_2$ shows the breakdown of solid carbonates in acidic conditions. The carbonate dissolution formula is reversible, but will proceed in the positive direction as temperatures increase. In susceptible areas, some closed or previously blocked karst pipes or cracks will open up under conditions of dissolution, and form new soil erosion channels. Dehydration of the soil will occur as the temperature

5    increases, and once runoff occurs or water levels rise, the dry soil will be removed, leading to erosion and disintegration as the sinkhole forms and collapses.

## 2.3 Correlation analysis

### 2.3.1 Sinkhole and drought peaks

Droughts in the USA can be divided into three basic, consistent peak periods: Phase i between 1965 and 1973, Phase ii

10   between 1973 and 1991 and Phase iii between 1991 and 2006. Sinkhole collapses in the USA can also be divided into three basic consistent peak periods: Phase i between 1968 and 1980, Phase ii between 1980 and 1993 and Phase iii between 1993 and 2006. From Fig. 3 it is evident that the peak time and trend of sinkhole collapse events and drought periods are quite consistent. To further investigate the relationship, the association can be quantified using curve fitting analysis.

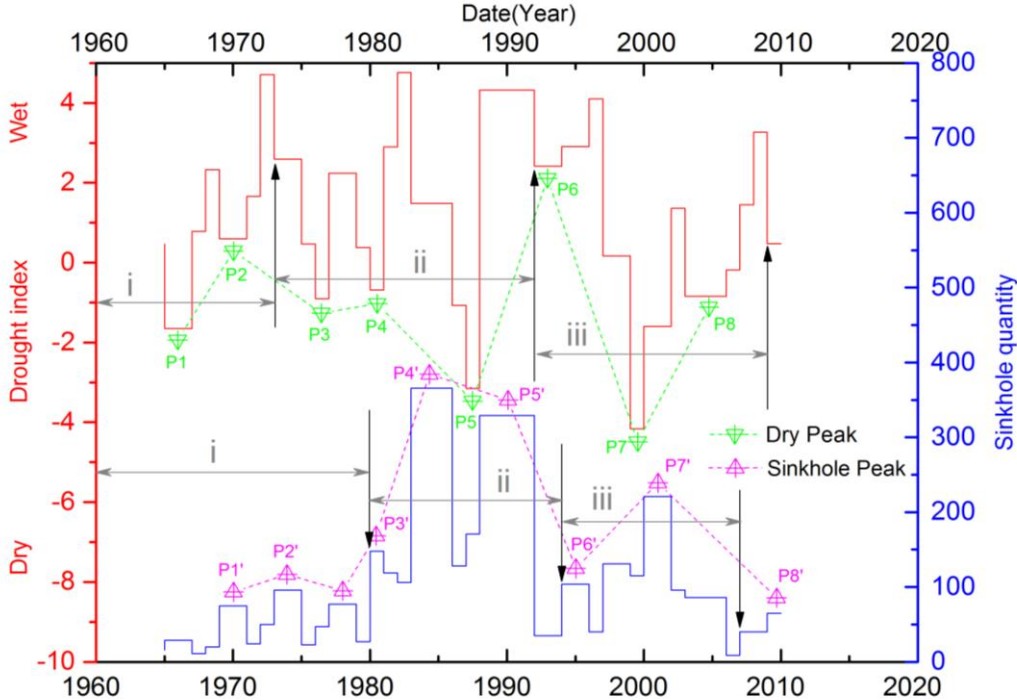

15   **Figure 3: Graphical illustration of the relationship between sinkhole quantity (blue) and drought periods (red) in Florida, USA. Note the eight peak "dry" and "sinkhole" periods shown (P1–P8) and the three corresponding phases (i, ii, iii) of highly consistent trends in sinkhole development and drought.**



### 2.3.2 Curve fitting

The curve of sinkhole collapse quantity and drought index can be fitted, as shown in Fig. 4, by Eq. (1).

$$\sqrt{((X - B)^2 + (Y - A)^2} - R = 0 \tag{1}$$

The algorithm is derived using the Quasi-Newton (Broyden Fletcher Goldfard Shanno (BFGS) and Universal Global(UG))

5   methods, where X is the drought index, Y is the number of sinkhole collapses and A, B and R are constant parameters. The correlation coefficient is 0.999. The other parameters are shown in Table 1.

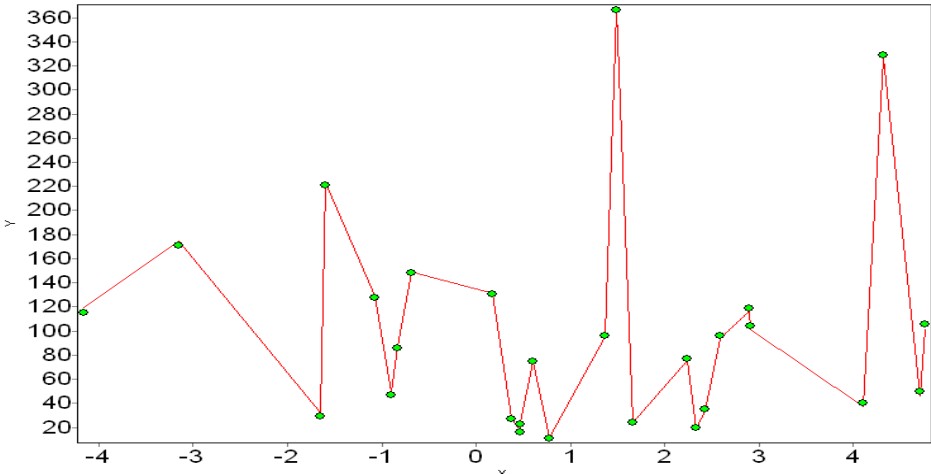

**Figure4: Fitted curves of sinkhole collapse quantity and drought index.**

**Table 1: Algorithm and parameters of the drought time curve.**

| Equation | | | Sqrt((X-B)^2 + (Y-A)^2) - R=0 |
|---|---|---|---|
| | A | 173.499 | Optimization: Quasi-Newton Method(BFGS) + Universal Global |
| | B | 8100.396 | Calculation End: Meet convergence criteria |
| | R | 8100.217 | Mean square error(RMSE): 2.17562755356349 |
| | | | Residual sum of squares (SSE): 127.800591799266 |
| Algorithm and parameters | | | Correlation coefficient(R): 0.999689237322014 |
| | | | The square of Correlation coefficient (R^2): 0.99937857121747 |
| | | | Determine the coefficient(DC): 0.999378190981562 |
| | | | Chi-Square coefficient: 0.958830651014433 |
| | | | F-Statistic: 40204.8713912301 |




## 3 Conclusions

There is a strong corresponding relationship between sinkhole quantity and drought index shown in Fig. 3, which demonstrates the link between global warming and increased development of sinkhole collapse events in Florida, USA. Eight peak points (P1–P8) within three peak drought periods correspond to sinkhole peak periods (P1'–P8'). The timing of
sinkhole formation lags behind the drought by two to four years, which is geologically sensible, given that water pumping and drawdown, along with soil runoff caused by rain, will take some time after the onset of drought before the sinkhole opens.

This is significant for use by government disaster reduction departments, or insurers, who may require forward-modeling of likely future events, such as sinkhole collapse following periods of drought. This will allow for controls of sinkhole collapse
to be established and to develop monitoring networks.

To clearly define the quantitative relationship between sinkhole collapse and drought, a curve fitting method was applied based on the optimization of Quasi-Newton (BFGS) and Universal Global methods. A prediction equation (Eq. 1) was also obtained according to the curve fitting.

It can be concluded that, if a drought period is forecast, the sinkhole quantity may also be forecast using the equation, and
similarly, areas in which quantities of sinkholes are increasing may be considered clear subjects of the impacts of global warming.

## Acknowledgements

This research was supported by the National Natural Science Foundation of China (41302255, 41402284), China Geological Survey Project (1212011220192). We thank Warwick Hastie, PhD, from Liwen Bianji, Edanz Group China
(www.liwenbianji.cn/ac), for editing the English text of a draft of this manuscript.

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
