# Peer review of "Global warming causes sinkhole collapse – Case study in Florida, USA"

_Natural Hazards and Earth System Sciences, 2018_

## Referee Comment (RC1) · Anonymous Referee #1 · 21 Feb 2018

While I found the topic of this manuscript interesting, there are some major problems with the research and the authors' conclusions which I have detailed below. There are two main issues: first, the authors do not seem to be aware of research conducted on this same problem in this same region of Florida. Second, I feel that the conclusions the authors reach are not supported by the actual reality of what is occurring in the study area. Consequently, I recommend rejecting this manuscript.

1. The very first thing I suggest is that the authors read this article as it pertains to their topic: Thornbush MJ (2017) Part 2: Spatial-Temporal Occurrences of Sinkholes as a Complex Geohazard in Florida, USA. J Geol Geophys 6:286. doi: 10.4172/2381-8719.1000286 There is also the book by Robert Brinkmann entitled Florida Sinkholes: Science and Policy.

2. Results are as good as the data. Do we know how well the data represents the actual occurrence of sinkholes? Many may occur in rural areas that may not require an insurance claim.

3. Page 3, Line 8: "sinkhole collapses recorded in Florida have three distinct peaks (Fig. 2)...". I only see two distinct peaks. Is there some statistical methods that the authors could use to justify these peaks as distinctive?

4. Page 3-4, last paragraph starting on page 3 needs references.

5. The section on sinkholes in Florida in the materials and methods really belongs in the introduction.

6. The authors do not have a results or discussion section, they include these in the materials and methods section.

7. The authors need to include a description of the drought index.

8. Figure 3 is difficult to interpret. Additionally, the authors classify P2 and P6 as dry periods yet they fall within the wet designation of the drought index.

9. The authors do not clearly state what the lags are between the droughts and the sinkhole collapse occurrences. From Figure 3 it appears that there are up to four years of lag between the drought and the sinkhole collapse. From my experience, sinkholes occur at the same period as droughts and which can drive the lowering of the water table which provides buoyant support to the "void roofs". For example, a cold snap in 2000s in central Florida required significant water withdrawals from the aquifer to prevent the freezing of crops. These withdrawals lower the water table leading to the formation of over 100 sinkholes forming in a matter of weeks, not four years.

10. The authors need to quantify/demonstrate changes in the water table which directly impacts the formation of sinkholes.

11. Figure 4 needs to be improved. The x and y axis need to be clearly labelled.

---

## Short Comment (SC1) · 8 Mar 2018

We are very grateful to the reviewers for their questions and suggestions on this article. As the experts put it, studying the relationship between global climate change and geological disasters is a global system science that requires multiple scholars to study from different perspectives. First, we carefully studied the articles provided by the experts.ïijĹThornbush MJ (2017) Part 2: Spatial-Temporal Occurrences of Sinkholes as a Complex Geohazard in Florida, USA. J Geol Geophys 6:286. doi: 10.4172/2381-8719.1000286ïijĽãĂĆWe are very interested in the content of the article, and the results of the article also prove the effect of temperature change on karst collapse. "The paper from this case study reveal a high incidence of sinkhole occurrence when temperatures are low and precipitation is also low in winter months (especially January). This suggests that temperature (rather than precipitation) may be the principal driving climatic factor, along with associated human impacts." We know that the formation of karst collapse is a complex process from dissolving limestone to surface pitsïijŇWe cannot deny that the ultimate collapse of the ground surface may be triggered by many factors such as human activities and earthquakes, However, during the formation of karst collapse, especially before the triggering of human activities, what are the factors at work should be the problem that we need to seriously study, which is also one of the research purposes of this article. Only Florida USA was chosen as a research example because of the good research base and database in this area. We revised the essay based on the opinions of experts and answered the specific questions as follows: 1. We have carefully studied the articles provided by experts. We are also very interested in the content of this article. The results of the article also seem to prove the impact and effect of temperature changes on karst collapse. We hope to have the opportunity to discuss with the author. 2. The data used in the article are based on official data from the USGS and publicly available data from the Institute of Florida sinkhole, not just data from insurance companies, which should be reliable. Even though these collapse data cannot contain all of Florida, the number of samples used and the macroscopic regularities of the responses can also support the viewpoint of this article from the perspective of mathematical statistics and scientific research. 3. There are indeed three peaks in Figure 2, the first peak period (1963-1976), the second peak period (1980-1993) and the third peak period (1994-2007). Since the amplitude of the first peak period is lower than the Second and third, so often overlooked. Some statistical methods proposed by experts to prove these peaks are unique. This is a good suggestion. We will conduct this research in future work. 4. We have added relevant references, on the third page 5,6,7. We adjusted the article structure according to the expert opinion, and explained the drought coefficient. 8,9. Global warming will lead to extreme climate changes such as droughts, torrential rains and frosts. It is also an important trigger factor for the collapse of human activities. The purpose of this paper is to analyze the relationship between global warming and karst collapse from a macro perspective, In the event of climatic anomalies, especially in the event of drought, the monitoring and prevention of karst collapse should be stepped up. The problem of collapse caused by other factors such as pumping in some areas and engineering construction should not be negated. 10ïijŐGroundwater change is an important trigger for the formation of karst collapse. However, this paper mainly discusses the relationship between global climate change and collapse. Therefore, no water level analysis is conducted and the author cannot obtain relevant data. 11.According to the expert opinion, the article has been modified.

Please also note the supplement to this comment:
https://www.nat-hazards-earth-syst-sci-discuss.net/nhess-2018-18/nhess-2018-18-SC1-supplement.pdf

―――――――――――――――――

**Supplement:**

[revised manuscript text omitted]

Alley, W.M., (1984) The Palmer Drought Severity Index: limitations and assumptions. Journal of Climate and Applied Meteorology, 23: 1100–1109.

Anikeev, A.V., Leonenko, M.V (2014) Forecast of sinkhole development caused by changes in hydrodynamic regime: Case study of Dzerzhinsk Karst Area. Water Resources 7:819–832.

Calbó, J., Sánchez-Lorenzo, A., Cunillera, J., Barrera-Escoda, A (2010) Projeccions i

Escenaris futurs J.E. Llebot (Ed.), 2n Informe sobre el Canvi Climatic a Catalunya, Grup d'Experts en Canvi Climatic de Catalunya. Generalitat de Catalunya i Institut d'Estudis Catalans, Barcelona. PP: 183-239.

Coe, J.A., Godt, J.W (2012) Review of approaches for assessing the impact of climate change on landslide hazards. In: Eberhardt, E., Froese, C., Turner, A.K., Leroueil, S.(Eds.), Landslides and Engineered Slopes, Protecting Society Through Improved Understanding: Proceedings 11th International and 2nd North American Symposium on Landslides and Engineered Slopes, Banff, Canada 1. Taylor & Francis Group, London. PP: 371–377.

Floor van den Heuvel., Stéphane Goyette., Kazi Rahman., Markus Stoffel (2016) Circulation patterns related to debris-flow triggering in the Zermatt valley in current and future climates. Geomorphology 272:127–136.

Gabriella Szépszó., Imke Lingemann., Bastian Klein., Mária Kovács (2014) Impact of climate change on hydrological conditions of Rhine and Upper Danube rivers based on the results of regional climate and hydrological models. Natural Hazards 1: 241–262.

Gutiérrez, F., Parise, M., De Waele, J., Jourde, H (2014) A review on natural and human-induced geohazards and impacts in karst. Earth-Science Reviews 138: 61-88.

Han Xiao.,Yong Je Kim., Boo Hyun Nam., Dingbao Wang (2016) Investigation of the impacts of local-scale hydrogeologic conditions on sinkhole occurrence in East-Central Florida, USA. Enviromental Earth sciences 75:1274.

IPCC, Climate Change 2013: The Physical Science Basis -Technical Summary (PDF). PP 89-90.

Jie Dong., Xue-feng Jia (2004) Possible impacts of global climate on natural disasters. Journal of Liaocheng Teachers College(Natural Science Edition) 2: 58-62 (in Chinese).

Mingtang Lei., Yongli Gao., Xiaozhen Jiang (2015) Current Status and Strategic Planning of Sinkhole Collapses in China. Engineering Geology for Society and Territory 5: 529-533.

Mulec, J., Prelovšek. M (2015) Freshwater biodissolution rates of limestone in the temperate climate of the Dinaric karst in Slovenia. Geomorphology 228:787-795.

Seneviratne, S.I., Nicholls, N., Easterling, D., Goodess, C.M., Kanae, S., Kossin, J., Luo, Y., Marengo, J., McInnes, K., Rahimi, M., Reichstein, M., Sorteberg, A., Vera, C., Zhang, X (2012) Changes in climate extremes and their impacts on the natural physical environment. In: Field, C.B., Barros, V., Stocker, T.F., Qin, D., Dokken, D.J., Ebi, K.L., Mastrandrea, M.D., Mach, K.J., Plattner, G.-K., Allen, S.K., Tignor, M., Midgley, P.M. (Eds.), Managing the Risks of Extreme Events and Disasters to Advance Climate Change Adaptation. A Special Report of Working Groups I and II of the Intergovernmental Panel on Climate Change (IPCC). Cambridge University Press, Cambridge, UK, and New York, NY, USA, pp. 109–230.

Stefano Luigi Gariano., Fausto Guzzetti (2016) Landslides in a changing climate. Earth-Science Reviews 162:227–252.

Thea Turkington., Alexandre Remaître., Janneke Ettema., Haydar Hussin., Cees van Westen (2016) Assessing debris flow activity in a changing climate. Climatic Change 137: 293–305.

Thornbush MJ (2017) Part 2: Spatial-Temporal Occurrences of Sinkholes as a Complex Geohazard in Florida, USA. Journal of Geology & Geophysics 6(3):286.

Wenxin Shi., Shuo Wang., Qianqian Yang (2010) Climate change and global warming. Reviews in Environmental Science and Bio/Technology 2: 99–102.

Yongming Lin., Haojun Deng., Kun Du., Loretta Rafay., Guangshuai Zhang., Jian Li., Can

Chen., Chengzhen Wu., Han Lin., Wei Yu., Hailan Fan., Yonggang Ge (2017) Combined effects of climate, restoration measures and slope position in change in soil chemical properties and nutrient loss across lands affected by the Wenchuan Earthquake in China. Science of The Total Environment 596:274-283.

Yuan Daoxian (1997) Modern karstology and global change study. Earth Science Frontiers 4 (1): 17-25 (In Chinese).

Zhou, Y.F., Tham, L.G., Yan, R.W.M., Xu. L (2014) The mechanism of soil failures along cracks subjected to water infiltration. Computers and Geotechnics 55: 330-341.

---

## Referee Comment (RC2) · Anonymous Referee #2 · 1 Apr 2018

This paper presents a study of the statistical correlation between drought periods and sinkhole occurrences in Florida over several decades. Overall, the topic is of interest and relevant to the field, but the study itself is lacking greatly in coherency, detail, and robust results to support the authors' claims; therefore, I suggest rejecting this manuscript until more data are supplied, along with a more detailed account of the data sources (e.g., the drought index) and statistical manipulations done to achieve the high correlation value presented herein.

Primarily, Figure 3 is difficult to interpret and it is difficult to tell how they authors achieved this high statistical correlation given the suggest lag, even using the methods presented, without doing more to adjust for frequency variability and potential errors in the data reporting (as well as other potential correlative causes, such as population

true

growth, reporting ability, and similar non-hydrologic or geologix influences).

The study lacks a thorough discussion and has overlooked or is missing several pertinent references on recent sinkhole studies in FL and elsewhere looking at their causes and prevalence. Drought alone doesn't account for increased overpumping, as population growth also plays a significant role in FL, especially in the past several decades in areas where sinkholes are most prevalent. The authors needs to provide a more thorough discussion with additional data considerations to rule out other causes. It would be possible to include hydrologic data as well, such as water table fluctuations in certain regions and PET (i.e. effective recharge), as these also could play a role and either support or contradict the findings of this study.

It is also unclear how a period is defined with respect to sinkhole occurrence, on which much of this study's outcomes are based and needs to be explained.

Overall, this paper has merit inits focus, but lacks in execution and in presenting a fully defendable dataset that lends insight to the true connection between climate change and sinkhole occurrences in FL without much more rigorous results and discussion sections.

---

## Author Comment (AC1) · 4 May 2018

We modified the article again. Sincerely, Yan Meng

Please also note the supplement to this comment:
https://www.nat-hazards-earth-syst-sci-discuss.net/nhess-2018-18/nhess-2018-18-AC1-supplement.pdf
* * *

---

## Author Comment (AC2) · 4 May 2018

First of all, we are very grateful to the review experts for their questions and suggestions. We know that the formation of karst collapse is a complex process from the dissolution of limestone to surface pits. We cannot deny that the occurrence of surface pits may be triggered by many factors such as human activities and earthquakes, but the formation of karst collapse In China, especially before the trigger of human activities, what factors are at work should be a problem that we need to seriously study. This is one of the objectives of this paper. Based on the opinions of experts, we have made major changes to the structure and content of the article. 1. Results and discussion sections were added. 2. A preliminary analysis of the relationship between karst collapse and global climate change in the Pearl River Delta region of China was

conducted. It was found that there is a similar relationship with Florida in the United States, thus providing further evidence for this study. 3. We modified the article map and added some necessary drawings. 4. We revise the ambiguity in the article. 5. The revision section was red in the article and the reference was added. Sincerely, Yan Meng

Please also note the supplement to this comment:
https://www.nat-hazards-earth-syst-sci-discuss.net/nhess-2018-18/nhess-2018-18-AC2-supplement.pdf

[Figure]

**Supplement:**

**Global warming causes increased sinkhole collapse– Cases studies in Florida, USA and the Pearl River Delta, China**

Yan Meng [a,b,*], Long Jia [b]

[a] School of environmental studies, China university of Geosciences, Wuhan, 430074,China

[b] Institute of Karst Geology, Chinese Academy of Geological Sciences/Karst Dynamics Laboratory, MLR & GZAR, CGS, Guilin, 541004,China

[*] Corresponding author: E-mail: sinkhole@163.com Tel.:+86 07737796682.

**Abstract**

The occurrence frequency and intensity of many natural geohazards, such as landslides, debris flows and earthquakes, have increased in response to global warming. However, the effects of such on development and spread of sinkholes has been largely overlooked. Most research shows that water pumping and related drawdown is the most important factor in sinkhole development, but in this paper evidence is presented which highlights the role played by global warming in causing more sinkholes. Cases were studied in Florida, USA and the Pearl River Delta of China. The results show that the four peak "dry" and the three highly phases (i, ii, iii) of sinkholes is closely related. A prediction equation was also obtained according to the curve fitting with the correlation coefficient is 0.99, which is of significance for studying the occurrence and prediction of other sinkhole collapse events and global warming on an international scale.

**Keywords**

Sinkhole; Drought index; Karst; Trend prediction; Curve fitting

**1. Introduction**

Global warming resulting from climate change has altered the occurrence frequency and intensity of many natural geohazards, including landslides, debris

flows and earthquakes (Calbó et al., 2010; Coe and Godt, 2012; Seneviratne et al., 2012; Gariano and Guzzetti, 2016; Heuvel, et al., 2016; Turkington, et al., 2016; Yongming Lin, et al., 2017). As an example of the mechanism for this, research has shown that 5%–10% of global permafrost will melt if global temperatures rise by 2°C, causing a significant increase in landslides and mudslides (Dong and Jia, 2004).

Sinkholes are a widespread type of geohazard, mainly distributed in the United States, China, Italy, Spain and Russia (Gutiérrez, et al., 2014; Lei, et al., 2015). The impact of climate change on sinkhole occurrence is expected, because rising temperatures will change natural hydrological processes (Gabriella Szépszó, et al., 2014), enhance dissolution of limestone (Mulec and Prelovšek, 2014) and promote soil failure (Zhou, et al., 2014). Recent reviews in the literature have shown that sinkhole hazards will probably intensify in the future as a result of climate change (Rogelio Linares, et al., 2017). The findings from the case (Thornbush, 2017) study reveal a high incidence of sinkhole occurrence when temperatures are low in winter months (especially January). This suggests that temperature (rather than precipitation) may be the principal driving climatic factor, along with associated human impacts. But the quantification of climatic impacts on sinkholes has been limited. This is largely because of a lack of long-term hydrological and climate data, and a lack of representative sinkhole inventories, inclusive of chronological information.

In this paper, the causal effects of global warming on sinkhole development and intensification are fully investigated using statistical analysis of sinkhole cases in the state of Florida, USA. There is a strong corresponding relationship between sinkhole increased after drought and drought indexes (Dry). A prediction equation was also obtained according to the curve fitting with the correlation coefficient is 0.99.

**2. Materials and methods**

*2.1. Drought index in Florida*

Global warming as a result of climate change is a quantifiable phenomenon (Shi et al., 2010; Gariano, et al., 2016; Turkington, et al., 2016), with a demonstrable increase

in global temperatures by ~0.57°C over the last century. It has been reported that the global surface temperature is likely to rise a further 0.3 to 1.7°C in the lowest emissions scenario during the 21st century, or by 2.6 to 4.8°C in the highest emissions scenario. It is an indisputable fact that global warming has caused drought (IPCC, 2013). Drought index is a measure of drought conditions and calculated based on rainfall, air temperature, and other meteorological factors (Keetch and Byram, 1968; Alley, 1984).

There are 9 droughts in Florida from 1960 to 2006 (1963, 1967, 1977, 1981, 1987, 1988, 2000, 2002, and 2006) (Fig.1). Drought was most severe in 2000, followed by 1987 and 1963.

[Figure]

**Fig. 1.** Drought indexes from 1960 to 2006 in Florida. The red line is dry, and the green line is wet

*2.2. Sinkhole collapse events in Florida*

In Florida, sinkhole collapse events are recorded in the Florida Subsidence Incident Report, authored by the Florida Geological Survey, which provides a primary publicly available sinkhole database. More than 2800 sinkholes have been reported in Florida since the 1950s, and 2767 of them were fully recorded between 1960 and 2006. The data recorded includes occurrence time, location, shape, dimensions, soil type, side slope, land use and land cover (Han, et al., 2016). Sinkhole claims were on the rise from 2006 to 2010. Sinkhole claims jumped from 2360 to 6694 in 2010, according to a 2010 report by the Florida Office of Insurance Regulation (Fig.2).

[Figure]

**Fig. 2.** Number of sinkholes from 1960 to 2006 in Florida

*2.3. Correlation analysis*

Sinkhole collapses in the USA from 2006 to 2010 can also be divided into three basic consistent peak periods: Phase i between 1963 and 1980, Phase ii between 1980 and 1992 and Phase iii between 1992 and 2006. From Fig. 3 it is evident that the peak time and trend of sinkhole collapse events and drought periods are quite consistent. To further investigate the relationship, the association can be quantified using curve fitting analysis.

[Figure]

**Fig.3.** Graphical illustration of the relationship between sinkhole quantity and drought in Florida, USA. Note the four peak "dry" and the three highly phases (i, ii, iii) of sinkholes is closely related.

It is very interesting that the relationship between sinkhole and global change in the Pearl River Delta of China is very similar to that of the Florida of USA. There is a

1  strong corresponding relationship between sinkhole quantity and drought index, and

2  they are consistent in the peak trend (Fig. 4).

[Figure]

4  **Fig.4.** Graphical illustration of the relationship between sinkhole quantity and drought

5  indexes in the Pearl River Delta of China. The green line is drought index average.

6  Note the four "dry" and "sinkhole" peaks are highly consistent trends.

7    The curve of sinkhole collapse quantity and drought indexes can be fitted, as

8  shown in Fig. 4, by Eq. (1).

9    $$\sqrt{((X-B)^2 + (Y-A)^2)} - R = 0 \qquad (1)$$

10    The algorithm is derived using the Quasi-Newton (Broyden Fletcher Goldfard

11  Shanno (BFGS) and Universal Global(UG)) methods, where $X$ is the drought index, $Y$

12  is the number of sinkhole collapses and $A$, $B$ and $R$ are constant parameters. The

13  correlation coefficient is 0.99. The other parameters are shown in Table 1.

[Figure]

14

15  **Fig. 4.** Fitted curves of increase in sinkhole after drought and drought index (Dry).
16    The blue line is target, the red line is calculated, the correlation coefficient is 0.99.

**Table 1** Algorithm and parameters of the drought time curve.

| Equation | | | Sqrt((X-B)^2 + (Y-A)^2) - R=0 |
|---|---|---|---|
| Algorithm and parameters | A | 173.499 | Optimization: Quasi-Newton Method(BFGS) + Universal Global |
| | B | 8100.396 | Calculation End: Meet convergence criteria |
| | R | 8100.217 | Mean square error(RMSE): 2.17562755356349 |
| | | | Residual sum of squares (SSE): 127.800591799266 |
| | | | Correlation coefficient(R): 0.999689237322014 |
| | | | The square of Correlation coefficient (R^2): 0.99937857121747 |
| | | | Determine the coefficient(DC): 0.999378190981562 |
| | | | Chi-Square coefficient: 0.958830651014433 |
| | | | F-Statistic: 40204.8713912301 |

**3. Results**

Macroscopically, the number and frequency of sinkholes increased with global warming from 1960 to 2006 in the Pearl River Delta of China and Florida, USA.

There is a strong corresponding relationship between sinkhole quantity and drought index shown in Fig.3 and Fig.4, which demonstrates the link between global warming and increased development of sinkhole collapse events. The four peak "dry" and the three highly phases (i, ii, iii) of sinkholes is closely related.

To clearly define the quantitative relationship between sinkhole increased after drought and drought index (Dry), a curve fitting method was applied based on the optimization of Quasi-Newton (BFGS) and Universal Global methods. A prediction equation (Eq. 1) was also obtained according to the curve fitting with the correlation coefficient is 0.99.

**4. Discussion**

[revised manuscript text omitted]

Stefano Luigi Gariano., Fausto Guzzetti (2016) Landslides in a changing climate. Earth-Science Reviews 162:227–252.

Thea Turkington., Alexandre Remaître., Janneke Ettema., Haydar Hussin., Cees van Westen (2016) Assessing debris flow activity in a changing climate. Climatic Change 137: 293–305.

Thornbush MJ (2017) Part 2: Spatial-Temporal Occurrences of Sinkholes as a Complex Geohazard in Florida, USA. Journal of Geology & Geophysics 6(3):286-292.

Wenxin Shi., Shuo Wang., Qianqian Yang (2010) Climate change and global warming. Reviews in Environmental Science and Bio/Technology 2: 99–102.

Yongming Lin., Haojun Deng., Kun Du., Loretta Rafay., Guangshuai Zhang., Jian Li., Can Chen., Chengzhen Wu., Han Lin., Wei Yu., Hailan Fan., Yonggang Ge (2017) Combined effects of climate, restoration measures and slope position in change in soil chemical properties and nutrient loss across lands affected by the Wenchuan Earthquake in China. Science of The Total Environment 596:274-283.

Yuan Daoxian (1997) Modern karstology and global change study. Earth Science Frontiers 4 (1): 17-25 (In Chinese).

Zhou, Y.F., Tham, L.G., Yan, R.W.M., Xu. L (2014) The mechanism of soil failures along cracks subjected to water infiltration. Computers and Geotechnics 55: 330-341.